# Trace Immunosensing of Multiple Neonicotinoid Insecticides by a Novel Broad-Specific Antibody Obtained from a Rational Screening Strategy

**DOI:** 10.3390/bios12090716

**Published:** 2022-09-03

**Authors:** Shasha Jiao, Yan Wang, Yunyun Chang, Pengyan Liu, Yang Chen, Yihua Liu, Guonian Zhu, Yirong Guo

**Affiliations:** 1Institute of Pesticide and Environmental Toxicology, Key Laboratory of Biology of Crop Pathogens and Insects of Zhejiang Province, Ministry of Agriculture Key Laboratory of Molecular Biology of Crop Pathogens and Insects, Zhejiang University, Hangzhou 310058, China; 2Research Institute of Subtropical Forestry, Chinese Academy of Forestry, Hangzhou 311400, China

**Keywords:** neonicotinoid pesticides, broad-specific mAb, immunochromatography, rapid detection

## Abstract

Residues of neonicotinoid pesticides have potential risks to food, environmental and biological safety. In this study, the hapten toward imidacloprid was adopted to gain antibodies. After molecular modeling, the electrostatic potentials of eight commonly-used neonicotinoid pesticides were individually calculated to analyze the structural similarity. Two representative compounds (imidacloprid and acetamiprid) with moderate similarity were rationally selected for hybridoma screening. Using this strategy, four clones of broad-specific monoclonal antibodies (mAbs) against multiple neonicotinoids were obtained, and the clone 6F11 exhibited the broadest spectrum to six neonicotinoid pesticides and two metabolites, with half-maximal inhibitory concentrations (IC_50_) ranging from 0.20 to 5.92 ng/mL. Then, the novel antibody gene was sequenced and successfully expressed in full-length IgG form using mammalian cells. Based on the sensitive recombinant antibody, a gold lateral-flow immunosensing strip assay was developed and it was qualified for rapid detection of imidacloprid, clothianidin or imidaclothiz residues in food samples.

## 1. Introduction

Neonicotinoids are currently the most widely used insecticides worldwide due to their high activity, good bioavailability, low toxicity to vertebrates and relative stabilities in the environment [1]. In recent years, their toxicity investigations have revealed that neonicotinoids display side-effects on pollinators [2], birds [3], and other non-target organisms [4]. The three most commonly detected neonicotinoids (clothianidin, imidacloprid and thiamethoxam) are classified as being highly toxic to bees (acute toxicity values, LD_50_, from oral ingestion are 1 to 5 ng/bee) [4]. Therefore, in April 2018, the European Union banned the use of these three neonicotinoid pesticides in open areas. To protect bees, the sale and use of five neonicotinoid pesticides (imidacloprid, acetamiprid, thiamethoxam, clothianidin and thiacloprid) have also been banned in France since September 2018. Therefore, it is essential to develop sensitive, reliable and rapid analytical methods for monitoring multi-neonicotinoid residues in food and environmental samples.

Liquid chromatography/tandem mass spectrometry is the dominant analytical technique for multi-residue determination of neonicotinoids in honey [5], environmental water [6] and agricultural products [7,8]. However, the application of this method is restricted due to the requirement of sophisticated instruments, skilled technicians and complex pretreatment procedures. Instead, immunoassays of different formats, such as enzyme-linked immunosorbent assay (ELISA), gold nanoparticle-based immunochromatographic strip (GNIS) assays or lateral-flow immunosensing tests, are feasible for rapid screening of agrochemicals. There is a trend to develop multi-analyte immunoassays (MAIAs) because they could simultaneously detect several targets in a single test. In a previous review, Li et al. discuss the various formats for the MAIA [9]. Although using multiple antibodies have been successfully applied to develop MAIAs, a more economical alternative is to raise a broad-specificity antibody that is able to recognize a class of the target compounds. Broad-specificity immunoassays had been developed for antibiotics [10], mycotoxins [11], veterinary drugs [12] and pesticides [13], and most of them can recognize more than three compounds from the same chemical families.

To date, many ELISAs have been reported for neonicotinoids and most of them were highly specific to a sole target pesticide, with some ELISAs showing cross-reactivity to other neonicotinoid analogues [14,15,16]. Among them, antibodies against imidacloprid have the most cross-reactivity to other neonicotinoids. The ELISAs thereof exhibited different cross-reactions to acetamiprid, thiacloprid or nitenpyram because the four neonicotinoids belong to the same structure type with a chloropyridine ring [14]. Moreover, an anti-imidacloprid monoclonal antibody (mAb) was applied to develop a time-resolved fluorescent nanobead-traced lateral-flow immunoassay, with the cross-reactivity to clothianidin (4.1%) and acetamiprid (1.1%) [17]. Additionally, an anti-imidaclothiz mAb-based ELISA exhibited high cross-reactivity to imidacloprid (91.7%) [18]. These previous studies inspired us that it would be possible to develop a broad-specific or generic mAb for more than five neonicotinoids.

The hapten design and hybridoma screening are the critical two steps in raising a broad-specificity mAb. In previous reports, the generation of broad-specificity antibody mostly relies on the design of generic hapten, which usually contain a “general structure” [9]. However, reports on hybridoma screening strategies for broad-specificity mAbs is rather rare. Normally, the screening process used a class of compounds as target analytes or just based on the single compound used to design the hapten. These traditional single- or all-screening methods either obtain broad-specificity antibodies usually by chance or occasionally or by labor-intensive and time-consuming methods. Once, a generic mAb against multiple phenothiazines was produced [19] using only acepromazine and chlorpromazine for hybridoma screening, but the method or the reason to select these two compounds was not explained in detail. Therefore, a rational strategy on selecting representative compounds for mAb test is highly valuable, which can greatly improve the screening efficiency and accuracy among thousands of hybridomas per cell fusion.

Computer-assisted molecular modeling is a powerful tool for hapten design and can be used to explain the cross-reactivity of antibodies [9,20,21]. Recently, there are a set of programs like Hyperchem, Gaussian and Sybyl that can provide quantitative information on chemical structures. For instance, molecular field overlapping was used to develop a molecular model of 27 different quinolones. The authors divided the 27 quinolones into three groups based on conformational similarity and exploited the hapten for selection according to the desired broad-specificity antibodies [22]. Therefore, we envisage using computer-assisted simulation to classify target analytes and select representative compounds for hybridoma mAb screening.

In the current work, the hapten toward imidacloprid was adopted for antibody generation. The experimental scheme was shown in Figure 1. According to the structural information of eight neonicotinoid pesticides, two representative compounds were rationally selected for hybridoma screening and several broad-specificity mAbs against four to six neonicotinoids were obtained. The novel mAb with the broadest selectivity and highest sensitivity was sequenced and successfully expressed in full-length IgG form using mammalian cells. Based on the broad-specific recombinant antibody (rAb), a sensitive GNIS assay was developed and it was qualified for rapid detection of multi-neonicotinoid residues in food samples.

## 2. Materials and Methods

### 2.1. Reagents and Materials

Standards of neonicotinoid insecticides and analogues were obtained from Agro-Environmental Protection Institute, Ministry of Agriculture (Tianjin, China). Bovine serum albumin (BSA), ovalbumin (OVA), pristine and rabbit anti-mouse IgG (whole molecule) secondary antibody conjugated with horseradish peroxidase (SecAb-HRP) were from Sigma-Aldrich (Madrid, Spain). Quick Antibody adjuvant (Mouse 5W), 3,3′,5,5′-Tetramethylbenzidine (TMB) and mouse antibody isotyping kit was supplied by Beijing Biodragon Immunotechnologies Co. (Beijing, China). Tween-20, gold chloride, trisodium citrate dihydrate and other chemical reagents were purchased from Shanghai Chemical Reagents Company (Shanghai, China). *E. coli* Trans1-T1 competent cells, pEASY-Blunt Zero vector and Trans DNA Marker were obtained from Beijing TransGen Biotech (Beijing, China). Restriction enzymes *Hind*III, *BamH*I and *EcoR*I were purchased from New England Biolabs (Ipswich, USA). RNAiso plus kit, SMARTer^®^ RACE 5′/3′ kit and agarose gel extraction kit were purchased from Takara Biotechnology (Tokyo, Japan). The HEK 293F cell line, vector pcDNA3.4, transfection reagent and serum free culture medium obtained from Biointron Biological Technology Co. (Shanghai, China). Ultra-pure water was acquired from a Milli-Q purification system (Millipore, Massachusetts, USA). Phosphate-buffered saline (PBS, 10 mM, pH 7.4) and carbonate-buffered saline (CBS, 50 mM, pH 9.5) were self-prepared. All other reagents were of analytical grade unless specified otherwise.

The mouse SP2/0 myeloma cells were obtained from the Cell Bank at the Chinese Academy of Sciences (Shanghai, China). MaxiSorp™ F96-well polystyrene microplates and different kinds of cell culture plates were purchased from Nalge Nunc International (Roskilde, Denmark). Glass-fiber conjugate pad, sample pad, and adsorbent pad were the products of Millipore (Massachusetts, USA). Nitrocellulose (NC) membrane was provided by Sartorius (Göttingen, Germany).

Standard stock solution of each neonicotinoid (1 mg/mL) was prepared in pure methanol and stored at 4 °C until use. Concerning the solubility of neonicotinoids (e.g., 610 mg/L of imidacloprid in water at 20 °C), their standard working solutions were freshly prepared by serial dilution with 1% (*v*:*v*) methanol-PBS.

### 2.2. Preparation of Hapten-Protein Conjugates

Hapten molecule for imidacloprid was synthesized as described in a previous report [23]. Using the active ester method, imidacloprid hapten was covalently attached to the lysine groups of BSA to serve as a representative immunogen. To prepare the coating antigen for immunoassays, imidacloprid hapten was conjugated with OVA via the mixed anhydride method with isobutyl chloroformate as the coupling reagent.

### 2.3. Computer-Assisted Simulation for Neonicotinoid Pesticides

The optimization and frequency of neonicotinoid molecules was calculated at the B3LYP level of density functional theory with the Gaussian 09 package (Wallingford, CT, USA). The 6-31G(d,p) basis set was used for all atoms. Dispersion correction was performed in order to increase the accuracy of the calculation. On the basis of confirming that all the molecules did not have virtual frequency, the single point energy was calculated at the level of B3LYP/6-311++G(d,p). The obtained wave function file was submitted to the Multiwfn software for quantitative molecular surface analysis [24]. The visualization of electrostatic potential (ESP) surface of molecules was displayed with Visual Molecular Dynamics 1.9.3(VMD 1.9.3) [25].

### 2.4. Immunization and mAb Production

Procedures for the immunization, cell fusion and clone selection were similar to those described in our previous publications [26], and indirect non-competitive and competitive ELISAs were sequentially used for hybridoma screening. The generated mAb was purified by protein G from the ascitic fluid and then stored at −20 °C for further analysis. A mouse antibody isotyping kit was used to determine the isotypes of the mAbs.

### 2.5. Full-Length RAb Production

#### 2.5.1. Cloning of mAb Variable Regions of Heavy Chains (VH) and Light Chains (VL)

Total RNA was isolated from 5  ×  10^6^ hybridoma cells using the RNAiso Plus Kit. Following the manufacturer’s instructions for SMARTer^®^ RACE 5′/3′ Kit, first-strand cDNA was synthesized and the antibody variable genes were amplified from cDNA via PCR (polymerase chain reaction) using the provided forward primer and subtype-specific primers (VH-Primer: CTCAATTTTCTTGTCCACCTTGGT; VL-Primer1: ACACTCAGCACGGGACAAACTCTTCTCCACAGT; VL-Primer2: ACACTCTGCAGGAGACAGACTCTTTTCCACAGT). A unique band at approximately 750 bp indicated a positive result. An Agarose Gel Extraction Kit was used to purify the VH and VL genes. 4 µL of each PCR product was cloned into the pEASY-Blunt Zero vector and then transform into the *E. coli* strain Trans1-T1 according to the manual. After overnight incubation at 37 °C, the plasmids were Sanger sequenced using the M13 universal primers. Each antibody variable region was submitted to NCBI with default parameters to determine percent identity to the reference sequences for light and heavy chains.

#### 2.5.2. Expression and Purification of the Full-Length rAb

An optimized vector containing signal peptide and constant region was generated from a customized pcDNA3.4. The VH and VL fragment obtained from hybridoma was cloned into pcDNA3.4 using an In-Fusion HD Cloning Kit, respectively. The plasmids were confirmed by Sanger sequencing and extracted using a Plasmid Midi Kit (QIAGEN, Duesseldorf, Germany).

Transient production in the HEK 293F suspension cells was performed as described [27]. Briefly, cells were prepared with viable cell density around 1.5  ×  10^6^ cells/mL and viability above 95 % in 180 mL serum-free culture medium in 500 mL Nunc Erlenmeyer flasks and incubated at 37 °C in 5% CO_2_ atmosphere and shaken at 130 rpm in an orbital shaker for 2 h. A total of 90 μg of the plasmids (VH:VL = 2:3) mixed with transfection reagent were pre-warmed for 15 min and then transiently transfected into HEK 293F. After 5 days of incubation, the supernatant was collected and purified by protein G. The size of the purified full-length rAb was identified by SDS-PAGE.

### 2.6. Characterization of Indirect Competitive ELISAs (icELISAs)

Checkerboard titration was carried out by icELISAs with serial dilutions of the antibody and coating antigen (imidacloprid-OVA). The combination of antibody and antigen dilutions that showed the highest inhibition by adding free imidacloprid was determined to be optimal for the following assays. The specificity and sensitivity were determined by icELISAs. Briefly, the mixture of standards with different concentrations and equal volume of antibody was added into the reaction wells of pre-coated plates. Standard curves were fitted into a four-parameter logarithmic equation: Y = A_2_ + [(A_1_ − A_2_)/1 + (X/X_0_)^p^], and the half-maximal inhibitory concentration (IC_50_, defined as the assay sensitivity, was used to calculate cross-reactivity rate(CR).

### 2.7. Development and Characterization of GNIS

The preparation method for the conjugate of rAb-gold nanoparticles was based on a previous report [27]. A colloid gold lateral-flow immunosensing strip consisted of three pads (glass fiber membrane, NC membrane and absorption membrane) using a one-sided adhesive polyvinyl chloride sheet as a support. The prepared imidaclothiz-OVA conjugate and goat anti-mouse IgG antibody were immobilized onto the NC membrane as the test line (T line) and control line (C line), respectively. For analysis, standard solutions at different concentrations were added to the 96-well microtiter plate, and then the strip was inserted into the well. After 15 min, the signal was observed using a portable reader (Helmen, Suzhou, China). Standard curves were gained by plotting the signal ratio of the T line to the C line (T/C) against the analyte concentration, and they were also fitted into the four-parameter logarithmic equation.

### 2.8. Sample Pretreatment and Spiked Recovery Test

Food samples of Chinese cabbage and honey were bought from supermarkets in Hangzhou, China, followed by homogenization. The samples were neonicotinoids-free, confirmed by Ultra-performance liquid chromatography-tandem mass spectrometry. For pretreatment, 2 g of each sample was extracted with 10 mL 0.01 M PBS by vortexing for 3 min and then filtered through 0.22 μm membrane, and the filtered solution was diluted to different folds with the same buffer before analysis. Matrix effects were evaluated by comparing standard curves in the matrix extracts with the curve prepared using matrix-free buffer. For spiked recovery test, each homogenized sample was separately added with imidacloprid, imidaclothiz and clothianidin at 12.5–100 ng/g and left overnight; the extracted solution was diluted to the optimized fold after the same pretreatment. Each analysis was performed in three replicates.

## 3. Results and Discussion

### 3.1. Screening Strategies for Broad-Specific mAbs

#### 3.1.1. Computer-Aided Selection of Representative Compounds

Structural similarity is often considered as the principle of immune hapten selection to induce broad-specific antibodies [28]. According to the previous reports on immunoassays for neonicotinoids, antibodies derived from imidacloprid hapten could be the most broad-selective, with diverse cross-reactivity to acetamiprid, thiacloprid, nitenpyram, imidaclothiz or clothianidin [14,17,18]. Thus, the hapten toward imidacloprid was adopted to gain broad-specific antibodies for neonicotinoids.

Similarly, the recognition spectrum or selectivity of anti-imidacloprid antibody is related to the structural similarity between imidacloprid and its analogues. To analyze the similarity degrees of 8 commonly-used neonicotinoid pesticides, Gaussian modeling was employed to calculate their ESP as shown in Figure 2. The cyan and yellow balls are the minimum and maximum values mapped to the molecular surface. According to the ESP, each compound is generally divided into three regions named a, b, and c (Figure 2B), and Types 1, 2 and 3 were used to indicate the similarity of each region (Figure 2C). Since the hapten toward imidacloprid was adopted as an immunogen, each region of imidacloprid is classified as Type 1.

By comparing the area corresponding to the ESP value of each region, the ESP difference relationship of eight compounds was obtained: (a) The terminal functional group is shown in Figure 2A as a dark red area, indicating this region has the large negative ESP distribution. This huge negative region is easily attracted by the potential positive region to form hydrogen bonds, so this region may be the key site for antibody binding. It could be observed that acetamiprid, thiacloprid and nitenpyram are quite different from the other five neonicotinoids (Figure 2D), and they are classified as Type 2. This is because the terminal structures of acetamiprid and thiacloprid are “=N–CN” and nitenpyram is “=C–NO_2_”, while those of the other 5 neonicotinoids are “=N–NO_2_”. (b) This part of the structure is chloropyridine, chlorothiazole or tetrahydrofuran in different compounds. Dinotefuran is classified as Type 3 because the area corresponding to each ESP value is the most different from the others (Figure 2E). The surface ESP distribution of the other seven compounds is relatively similar, but they have obvious differences in the ESP value of 9.5 kcal/mol. Therefore, imidacloprid and thiamethoxam are classified as Type 1, and the other five compounds are classified as Type 2. (c) The structure of this region shows the positive potential (dark blue). It could be seen from the histogram in Figure 2F that nitenpyram and dinotefuran are significantly different from other compounds in the area of each ESP value, so they are classified as Type 3. At several points with a high area ratio, such as 9.5, 16.5 and 23.5 kcal/mol, acetamiprid and thiamethoxam showed moderate differences, so acetamiprid and thiamethoxam are classified as Type 2, and the rest belong to Type 1.

To sum up, imidacloprid was first determined for use as the test analyte as the immunogen was derived from its structure. Compounds with highly similar conformations must have the similar binding mechanism, while compounds with large differences may not be recognized by the same antibody. Hence, acetamiprid, which has moderate similarity to imidacloprid in all three regions, was selected as another representative compound for hybridoma screening, and we predicted that the broad-specific mAb recognizing both imidacloprid and acetamiprid must have cross-reactivity to other neonicotinoids, such as thiacloprid, nitenpyram, imidaclothiz, clothianidin or thiamethoxam, based on the molecular modeling.

#### 3.1.2. Hybridoma Screening by Dual-Target icELISAs

After cell fusion and the primary screening by non-competitive ELISAs, 37 positive wells of hybridomas giving the absorbance values higher than 0.5 were picked out and filled with fresh culture medium. On the next day, they were quickly rechecked by icELISAs to qualitatively test the ability of recognizing both imidacloprid and acetamiprid. It was found nine clones showed strong inhibition rate (>50%) by imidacloprid at 1 µg/mL, but their inhibition by acetamiprid at 1 µg/mL varied from low (<10%), to middle (10–50%), to high (>50%) levels. As shown in Figure 3, clones 6F11, 1A5, 4B1 and 5C3 showed strong recognition to both imidacloprid and acetamiprid. Therefore, the top four broad-specific clones were selected and screened with imidacloprid and acetamiprid until stable hybridoma lines were obtained. Then the culture supernatants were collected for characterization of the four kinds of broad-specific mAbs.

Using non-competitive iELISAs with coating antigen (imidacloprid-OVA at 10 µg/mL), titers of supernatants were first determined and they were in the range of 40–80. Moreover, sensitivity (IC_50_) and CR to eight neonicotinoids were assessed by icELISAs. Judging from the CR values (Table 1), the broad-specificity of four mAbs followed this order: 6F11 > 1A5 > 5C3 > 4B1. The former three mAbs exhibited different sensitivity to six neonicotinoids including imidacloprid, imidaclothiz, clothianidin, acetamiprid, thiacloprid and nitenpyram, without obvious CR to dinotefuran and thiamethoxam. Comparatively speaking, mAb 4B1 was less sensitive to imidacloprid than other mAbs, and it exhibited no CR to nitenpyram, dinotefuran and thiamethoxam. Generally, the experiment results were in agreement with the computational prediction mentioned above, except that thiamethoxam could not be recognized by all the broad-specific mAbs. Thus, to the best of our knowledge, the newly-developed mAbs are found to be more broad-specific than other previously-reported antibodies against neonicotinoids [14,17,18]. Nevertheless, these data were from the hybridoma cell supernatant and further characterization of purified mAbs should be conducted in a following study.

### 3.2. Antibody Sequencing and Expression

Hybridoma-dependent ascites mAbs are usually used for pesticide detection. However, hybridoma cells sometimes have problems with variable region gene diversity, mutation or even loss, resulting in unstable antibodies [16,27]. On the other hand, compared to mAbs, rAbs are relatively simple to prepare without experimental animals and hybridoma, eliminating ethical and animal welfare concerns [29,30]. Therefore, once the valuable mAb clone is obtained, it is highly necessary to identify the antibody variable region (VR) sequence for immortalization or reproducibility. With the development of DNA recombination technology, the functional sequences can be expressed in different systems. Mammalian cell expression has the advantages of protein folding and solubility, post-translational modifications and secretion apparatuses, resulting in the high-quality products [29]. Therefore, the hybridoma clone 6F11 with the broadest selectivity and highest sensitivity was selected for generation of full-length rAb by mammalian expression system. The 6F11 mAb was IgG1 isotype and λ light chain. The total RNA was isolated from hybridoma cells, and a unique band at approximately 750 bp can be clearly seen in Fig.S1A after the PCR reaction. Five days after transient expression, cell supernatants were harvested and measured by icELISAs for analysis of antibody activity. Then the rAb was purified by protein G and analyzed with SDS-PAGE. Under reducing conditions, two bands at approximately 25 kDa (light chain) and 50 kDa (heavy chain) were revealed (Appendix A).

### 3.3. Evaluation the Sensitivity and Specificity of 6F11 mAb and Full-Length rAb

Under the optimal working concentrations of 10 µg/mL for imidacloprid-OVA, icELISAs with high sensitivity to neonicotinoid pesticides and metabolites were established. Results of the icELISA showed that the mAb and rAb have similar characteristics. As shown in Figure 4, both 6F11-mAb (Figure 4A) and 6F11-rAb (Figure 4B) exhibited high sensitivity against the six neonicotinoid pesticides and two metabolites, with IC_50_ ranging from 0.20 to 9.04 ng/mL (Table 2). It was clear that the assay showed higher sensitivity to imidacloprid, imidaclothiz, clothianidin, 6-Cl-PMNI and N-desmethyl thiamethoxam than thiacloprid, acetamiprid and nitenpyram.

Notably, the sensitivity of the antibody to the compound correlates strongly with the similarity of the compounds based on ESP. This means that ESP analysis of compounds has the potential to predict the recognition performance of antibodies with different compounds. There was also an inconsistency among the results; that is, thiamethoxam is highly similar in structure to imidacloprid, but the antibody cannot recognize thiamethoxam. We speculate that it is due to the potential steric hindrance between the methyl group in the region c of thiamethoxam and the surrounding residues because the antibody has a very high affinity against N-desmethyl thiamethoxam.

Another important point is that the imidacloprid hapten used in this study was currently the most used one in the preparation of antibodies against imidacloprid, but those antibodies only showed cross-reactivity with two or three other neonicotinoids [14,17,31]. We noted that those previously-reported antibodies were typically selected by using a single analyte for hybridoma screening based on ic-ELISAs, while this study improved the traditional screening modes. By using molecular modeling and surface ESP calculated by Gaussian modeling, two representative compounds were determined to use in dual-target icELISA for hybridoma screening. Thus, a high-affinity and broad-specificity mAb (clone 6F11) against eight neonicotinoids was newly developed, which is the broadest-spectrum antibody against neonicotinoids so far. In fact, the dual-target icELISA has been mentioned in a previous report [19], but the researchers did not explain the relevant experimental design in detail.

Above all, the present work confirms the feasibility of the computer-aided method and provides a rational screening strategy for the development of targeted antibodies. Since the main purpose of this study is to screen broad-specificity antibodies, two representative compounds with moderate similarity were selected after computational analysis. On the contrary, if the goal is to screen highly specific antibodies, it is more appropriate to select compounds with high conformational similarity as co-analytes.

### 3.4. Analysis of Food Samples by GNIS Test

The 6F11-rAb was used to establish a rapid and reliable GNIS assay, and the limit detection of T-line disappearance was 5 ng/mL for imidacloprid, clothianidin and imidaclothiz and over 100 ng/mL for thiacloprid, acetamiprid and nitenpyram, judged by normal visual detection (Figure 5). With a strip reader, the assay showed that the IC_50_ values for the detection of imidacloprid, clothianidin and imidaclothiz were 0.9, 0.5 and 0.7 ng/mL, respectively. Therefore, imidacloprid, clothianidin and imidaclothiz, with better sensitivity, were further determined by spiking experiment. The results of the matrix effect showed that the Chinese cabbage and robinia honey samples need to be diluted 10 times and 5 times, respectively (Appendix A). For spiked levels at 12.5–100 ng/g in Chinese cabbage and honey samples, the mean recoveries and relative standard deviations (RSDs) were in the range of 76.0–128.4% and 3.4–11.0%, respectively (Table 3). In general, the above results indicated that the strip immunosensor based on the 6F11-rAb could be applied to detect imidacloprid, imidaclothiz or clothianidin residues in food samples.

## 4. Conclusions

In this study, according to molecular modeling and surface ESP calculated by Gaussian modeling, two representative compounds (imidacloprid and acetamiprid) with moderate similarity were rationally selected to use in dual-target icELISA for hybridoma screening. Using this strategy, four clones of broad-specific mAbs against multiple neonicotinoids were obtained, and the clone 6F11 exhibited the broadest spectrum to eight neonicotinoids compounds: imidacloprid, imidaclothiz, clothianidin, thiacloprid, acetamiprid, nitenpyram, 6-Cl-PMNI and N-desmethyl thiamethoxam, with IC_50_ ranging from 0.20 to 5.92 ng/mL. Then, the novel antibody gene was sequenced and successfully expressed in full-length IgG form using mammalian cells. Based on the broad-specific rAb, a sensitive GNIS assay was developed, with the visual detection limit of 5 ng/mL for imidacloprid, imidaclothiz and clothianidin. In short, this study provides a computer-aided rational screening strategy for efficient development of broad-specificity antibodies, and the corresponding rAb can be permanently produced and used in diverse immunosensors for rapid detection of multi-neonicotinoid residues in agricultural or environmental samples.

## Figures and Tables

**Figure 1 biosensors-12-00716-f001:**
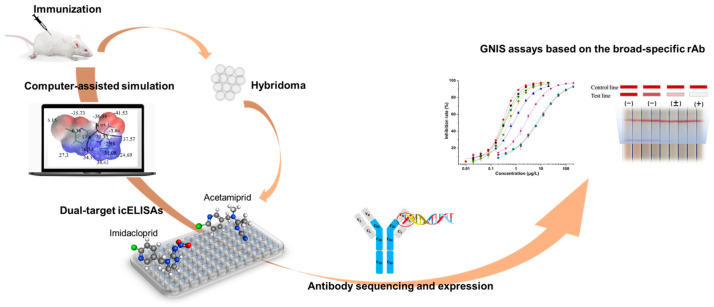
Scheme of the development of a novel broad-specific antibody and the immunosensing strip for 8 neonicotinoids.

**Figure 2 biosensors-12-00716-f002:**
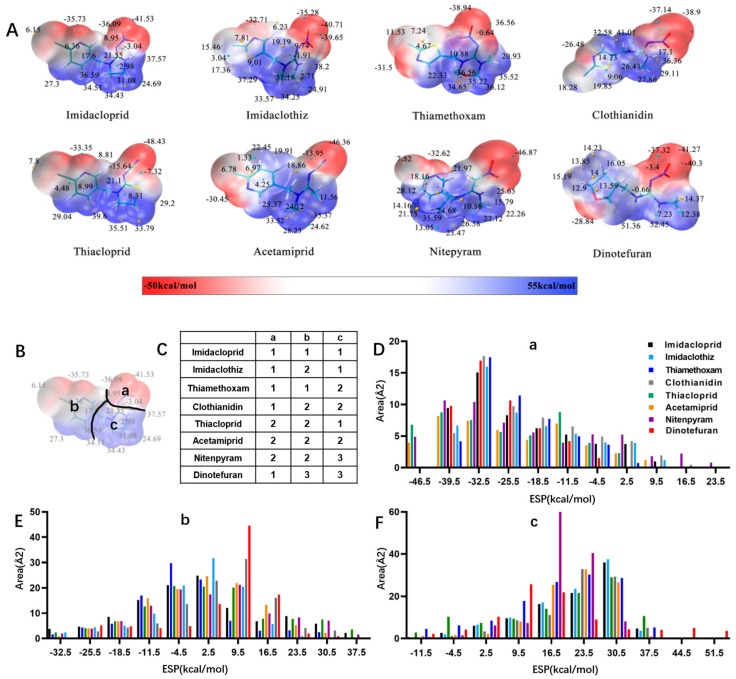
ESP analysis of 8 neonicotinoid pesticides. (**A**) ESP distribution on the surface of 8 neonicotinoid pesticides. The cyan and yellow balls are the minimum and maximum values mapped to the molecular surface. (**B**) Compound region division based on ESP characteristics. (**C**) Similarity of each region, indicated by Types 1, 2 and 3. (**D**–**F**) Surface area in each ESP range of three regions (a, b, and c).

**Figure 3 biosensors-12-00716-f003:**
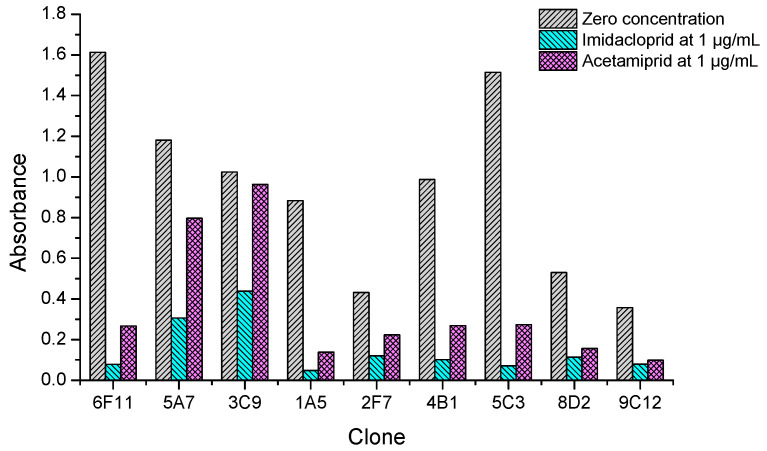
Inhibition tests of the positive clones by dual-target icELISAs coated with imidacloprid-OVA at 10 µg/mL. Data are the mean values of two replicates.

**Figure 4 biosensors-12-00716-f004:**
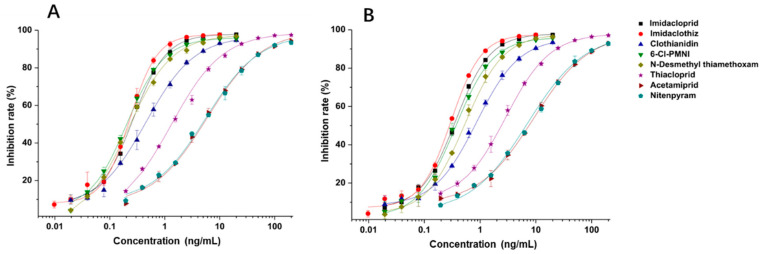
Standard curves for the detection of neonicotinoid pesticides and metabolites by icELISAs based on 6F11-mAb (**A**) and 6F11-rAb (**B**). The error bars represent the standard deviations of the data points (*n* = 3).

**Figure 5 biosensors-12-00716-f005:**
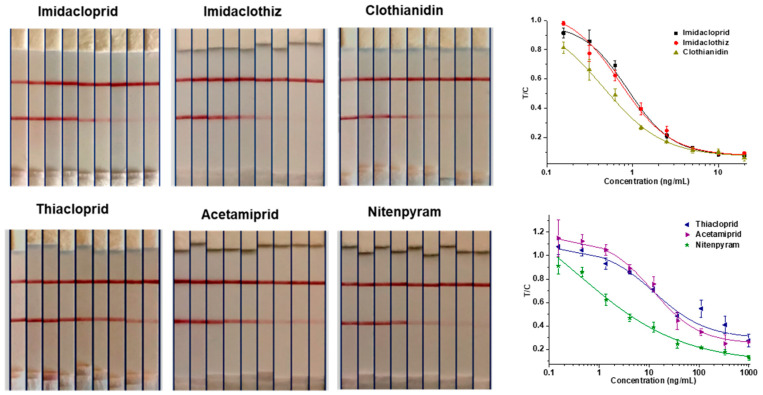
GNIS test for serial concentrations of 6 neonicotinoid pesticides (imidacloprid, imidaclothiz and clothianidin from left to right: 0, 0.16, 0.31, 0.63, 1.25, 2.5, 5, 10, 20 ng/mL; thiacloprid, acetamiprid and nitenpyram from left to right: 0, 0.46, 0.37, 4.12, 12.35, 37.04, 111.11, 333.33, 1000 ng/mL). Standard curves represent the ratio of the T-line to the C-line (T/C) against analyte concentration. The error bars represent the standard deviations of the data points (*n* = 3).

**Table 1 biosensors-12-00716-t001:** Sensitivity (IC_50_, ng/mL) and CR (%) of four mAbs toward 8 neonicotinoid pesticides.

Neonicotinoids	mAb 1A5		mAb 4B1		mAb 5C3		mAb 6F11	
	IC_50_	CR	IC_50_	CR	IC_50_	CR	IC_50_	CR
Imidacloprid	1.14	100.0	2.90	100.0	0.24	100.0	0.59	100.0
Imidaclothiz	0.90	126.7	41.09	7.1	2.79	8.6	0.48	122.9
Clothianidin	1.80	63.3	59.41	4.9	1.19	20.2	1.43	41.3
Thiacloprid	42.91	2.7	28.37	10.2	13.39	1.8	2.78	21.2
Acetamiprid	43.34	2.6	92.77	3.1	58.52	0.4	6.24	9.5
Nitenpyram	31.19	3.7	>1000	<0.1	30.66	0.8	7.16	8.2
Dinotefuran	>1000	<0.1	>1000	<0.1	>1000	<0.1	>1000	<0.1
Thiamethoxam	>1000	<0.1	>1000	<0.1	>1000	<0.1	>1000	<0.1

**Table 2 biosensors-12-00716-t002:** Selectivity of 6F11-mAb and 6F11-rAb toward neonicotinoid pesticides and metabolites.

Neonicotinoids	6F11-mAb		6F11-rAb	
	IC_50_ (ng/mL)	CR (%)	IC_50_ (ng/mL)	CR (%)
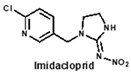	0.27	100.0	0.36	100.0
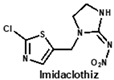	0.23	115.1	0.30	119.8
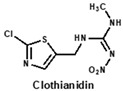	0.46	58.3	0.86	41.7
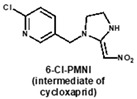	0.20	132.9	0.36	100.6
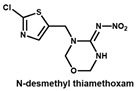	0.21	124.8	0.53	67.8
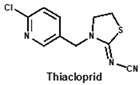	1.30	20.6	2.89	12.5
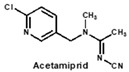	5.92	4.5	9.04	4.0
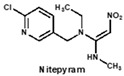	5.57	4.8	7.07	5.1
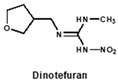	>1000	<0.1	>1000	<0.1
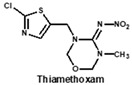	>1000	<0.1	>1000	<0.1

**Table 3 biosensors-12-00716-t003:** Average recoveries of sample spiked with 3 neonicotinoid pesticides based on GNIS assays (*n* = 3).

Sample	Pesticide	Spiked Level (ng/g)	Detected Concentration (ng/g)	Average Recovery (%)	RSD (%)
Chinese cabbage	Imidacloprid	25	21.0 ± 1.9	84.0	9.1
		50	64.2 ± 2.5	128.4	3.9
		100	124.0 ± 13.7	124.0	11.0
	Imidaclothiz	25	23.2 ± 1.7	92.8	7.5
		50	42.9 ± 4.4	85.8	10.4
		100	123.0 ± 5.7	123.0	4.0
	Clothianidin	25	26.5 ± 1.4	106.0	5.5
		50	58.6 ± 5.5	117.2	9.4
		100	114.0 ± 7.8	114.0	6.8
Robinia honey	Imidacloprid	12.5	13.4 ± 1.1	107.2	8.5
		25	23.9 ± 2.4	95.4	9.9
		50	61.5 ± 3.0	123.0	4.8
	Imidaclothiz	12.5	12.0 ± 0.4	95.6	3.4
		25	26.5 ± 1.2	105.8	4.6
		50	59.5 ± 2.4	119.0	4.0
	Clothianidin	12.5	10.2 ± 0.9	81.4	8.7
		25	19.0 ± 0.7	76.0	3.6
		50	55.0 ± 3.5	110.0	6.3

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
