# Peer review of "Trace Immunosensing of Multiple Neonicotinoid Insecticides by a Novel Broad-Specific Antibody Obtained from a Rational Screening Strategy"

_biosensors, 2022, doi:10.3390/bios12090716_

Round 1

Reviewer 1 Report

The authors described an rational hapten design with computer assistance, and prepared monoclonal antibody to neonicotinoid, based on the cell obtained, recombinant antibody with full length IgG expressed by mallamian cell, the antibody demostrated broad specificity to the analogues of neonicotinoid with good sensitivity, and the proposed method was well evaluated, all the work is systimatically carried out. The topic is novel, and result is supported enough with data, the lateral flow immunochramotography established is valuable for screening pesticide residue. Just a few question for authors to clearify and enhance the readability.

1. The novelty of this investigation may be focused and emphasized in the title. 

2. The monoclonal antibody could be used for assay development, however, the recombinant one was used in the strip, please add the analysis or discussion about the the necessarity, advantage and disadvantage of recombinant one, compared to traditional antibodies.

3. Add some comparison with other similar investigation.

Reviewer 2 Report

Authors covered all the background in the introduction and detailed method part, as well. Results are sufficient but there are a few flow needs to be address:

-Page 7 - Caption for Figure 2: It says "Data were the mean values of two replicates." However, in the page 5, it says "Each analysis was performed in three replicates." please correct the inconsistency by performing analysis in Figure 2 one more time so that you can report three replicated as you claimed in Page 5. 

-Novelty statement is missing. Author should highlight why this working is better/ more advanced than similar reported works.  

Reviewer 3 Report

The article is of very high quality, in my opinion it could be published with minimal changes resulting from normal editorial changes. Neverthless, I recommend (or suggest) adding 2 notes to the text, which could hopefully increase the interest of readers:

1. In the introduction of the paper, the authors discuss the hazards and risks of neonicotinoid pesticides (and cite relevant sources), but nowhere mention hard numerical data (e.g., lethal or harmful doses and concentrations) that would imply high immuno-sensor.

2. I think the text would be clearer it if included a simple picture (block diagram) of the function of the immunosensing strip.

Otherwise, I wish the authors well in their future scientific work. 
